# Environmental Attitudes in Trainee Teachers in Primary Education. The Future of Biodiversity Preservation and Environmental Pollution

**DOI:** 10.3390/ijerph16030362

**Published:** 2019-01-28

**Authors:** Inmaculada Aznar-Díaz, Francisco-Javier Hinojo-Lucena, María-Pilar Cáceres-Reche, Juan-Manuel Trujillo-Torres, José-María Romero-Rodríguez

**Affiliations:** Department of Didactics and School Organization, University of Granada, 18071 Granada, Spain; iaznar@ugr.es (I.A.-D.); fhinojo@ugr.es (F.-J.H.-L.); caceres@ugr.es (M.-P.C.-R.); jttorres@ugr.es (J.-M.T.-T.)

**Keywords:** environmental attitudes, environmental education, primary education, biodiversity conservation, environmental pollution, teaching, Spain

## Abstract

The environmental challenges of the twenty-first century are a consequence of the impact that human beings have on nature. Because of this, environmental attitudes are paramount in establishing effective measures regarding both biodiversity preservation and environmental pollution. Therefore, the main goal of this paper has been evaluating the environmental attitudes of future primary education teachers, considering their responsibility for teaching the new generations of citizens. A quantitative methodology has been applied to describe the reality observed. The data collection instrument used was the scale of environmental attitudes towards specific problems; this scale was applied to a sample of 307 students of the degree in Primary Education at the University of Granada. More specifically, the subscales corresponding to pollution, biodiversity, natural spaces and recycling were analysed. Results reveal a high level of environmental attitudes in future primary education teachers; moreover, there is a positive interdependence among the different subscales. Finally, it is of great importance to evaluate the environmental attitudes of future primary education teachers, given that the future of biodiversity preservation and environmental pollution are in the hands of new generations.

## 1. Introduction

Environmental degradation has increased over recent decades, resulting in a worldwide issue [1,2]. This has had an impact on society, where certain public institutions and private companies (Spanish Office of Climate Change, United Nations, Goldman Sachs, Google, Apple, Microsoft) have put in place a number of policies and programmes at microeconomic and macroeconomic levels with the aim of slowing down an environmental disaster [3,4]. Some measures have also been taken individually. Specifically, recycling and waste reduction are the main actions at home, given that the status of natural settings has a direct impact on the health of human beings [5].

In its latest report, the World Wildlife Fund (WWF) highlights the major environmental challenges of the twenty-first century, including environmental pollution, deforestation of natural spaces and loss of biodiversity [6]. 

Some of the ecological disasters caused by humans (the Chernobyl nuclear disaster, the desertification of the Aral Sea, the oil spills in the Niger Delta, the Fukushima nuclear disaster, the toxic cloud over Seveso, the oil spills in the Gulf of Mexico, the Prestige oil spill), have led to catastrophic consequences for the environment, and for human, plant and animal life, even accelerating deforestation and the loss of diversity in some cases. On the other hand, the use of fossil fuels (coal and crude oil), together with toxic gas emissions into the atmosphere have played a negative role in the increase of the Earth’s temperatures [7]. Hence, it is necessary to reduce toxic emissions and leave fuels like coal behind and start using renewable energies [8].

Pollution caused by humans undoubtedly affects both natural spaces and biodiversity, which results in drastic consequences for ecosystems, because it produces not only a loss in species balance but also the degradation of natural habitats [9,10,11]. 

The current situation and the different ecological disasters have underscored citizens’ environmental attitudes, which refers to the set of facts and images regarding the environment that the human mind stores and which affect our decisions [12]. Hence, there is a continuous cycle of the human impact on nature and how that impact subsequently affects our health and our environmental attitudes (Figure 1).

On the other hand, recycling is one of the measures that have been far-reaching in terms of citizens’ attitudes. This is the process that aims to turn waste into a new product to allow for reuse [13]. As examples, we have the recycling of cardboard, organic matter, and glass [14], and even the water we use [15]. Nevertheless, some countries like Spain only recycle 29% of municipal waste, a figure which is very far from the goal set by the European Union for 2020 [16].

In this regard, the role of new generations is paramount for the future preservation of and care for the environment [17,18,19]. Environmental education may therefore be the answer to environmental issues [20,21,22,23,24]. Nonetheless, there are shortcomings and constraints when addressing environmental education in educational centres [25,26]—in the first instance, teachers’ own awareness of environmental issues, and secondly, the poor environmental training they have received [9,27,28,29].

Universities have tried to implement environmental education as a cross-disciplinary subject in the official syllabus; it was the establishment of the Network for the Curriculum’s Environmental of Higher Education in Spain (ACES Network) [28]. Despite this fact, insufficient levels of environmental literacy are seen in future teachers [30]. Indeed, it is necessary to give trainee teachers in primary education environmental education, so that they can later share their knowledge with future generations [12,31].

Concerning previous studies on the environmental attitudes of students from different educational levels and areas of knowledge, a rich landscape is shown in the perception of environmental issues. Espino, Olaguez, and Davizon [32] studied the environmental perception in university students of mechatronic engineering at the Polytechnic University of Sinaloa (Mexico). It was found that the main issues in their locations are pollution and the destruction of natural resources. Ramírez [33] focused on the perception of Colombian university students from different institutions with the aim of gathering information regarding environmental issues. He concludes that their primary concern is water and air pollution, together with a poor management of solid waste. Soga, Gaston, Yamaura, Kurisu, and Hanaki [34] analysed the effects of interacting with the environment on children’s environmental awareness in Tokyo. It was shown that children who experience encounters with nature more frequently develop greater ecological awareness and a disposition to preserve biodiversity. Gädicke, Ibarra, and Osses [17] analysed environmental perceptions in Secondary School students in Temuco (Chile) and possible gender differences. These authors remark that students have knowledge of environmental challenges and that they perceive them as serious for nature and human beings. The existence of statistically significant differences around gender was not confirmed. Álvarez, Sureda, and Comas [30] prepared an assessment on the environmental competencies of teachers of Primary Education Teaching at the University of the Balearic Islands (Spain). Students lacked adequate environmental competencies for teaching their future students. Pérez-Franco, Pro-Bueno, and Pérez-Manzano [35] analysed the environmental attitudes of secondary education students in the Region of Murcia (Spain). Results show a positive attitude towards environmental concern. Their work also asserts that there are gender differences in favour of women. Lastly, Brito, Oliveira, and Silva [36] carried out a study on environmental awareness of a group of university students from Marajó (Brazil). They explain that students have a good knowledge of environmental practices, though certain difficulties in applying their knowledge to real life are shown.

Overall, it is crucial to study environmental attitudes if we want to understand the links generated between environmental preservation and current global challenges [32,37]. It is also important to successfully implement environmental education in primary education classrooms, and to pass on environmental values to future generations that ensure care for the environment and biodiversity preservation. Therefore, this paper has a number of aims: (i) analysing future primary education teachers’ environmental attitudes; (ii) learning whether there are statistically significant differences based on gender; (iii) establishing correlations between the study subscales (recycling, pollution, natural spaces, and biodiversity) through structural equation modelling (SEM).

## 2. Method 

Based on different studies that analyse environmental awareness or attitudes [17,30,32,33,34,35,36], we decided to follow the same methodological approach. In this sense, the research has been carried out from a quantitative perspective, with the aim of describing the reality observed and of gathering empirically testable data [38]. This has allowed quantifying of participants’ answers in a numerical way. Furthermore, statistic-descriptive values, a comparison between groups, and the correlation among latent variables have been established. 

In parallel, according to the aims of the study, the following questions for the research were posed:▪What is the environmental attitude of future primary education teachers towards pollution and its repercussions for biodiversity and natural spaces?▪What are future primary education teachers’ attitudes towards recycling?▪Are there statistically significant differences in the environmental attitudes according to gender?▪Does the level of attitudes towards issues related to pollution, biodiversity preservation and natural spaces have an influence on having a positive predisposition towards recycling? 

### 2.1. Participants

The study population was focused on second-year students studying for a Primary Education degree at the University of Granada in the 2018–2019 academic year. To determine the study sample, cluster probability sampling was used [39]. Clusters were predetermined based on the group each student belonged to. The degree of Primary Education at the University of Granada has eight groups in the second year (*n* = 515). Therefore, five groups were randomly selected to reach a significant sample size (*n* = 307), which was determined with a 99% confidence interval and a 5% margin of error.

Regarding participants’ gender and age, 74.3% of them are women (*n* = 228) and 25.7% are men (*n* = 79). The majority percentage of women is quite common among university degrees within the education sphere [40]. The age range is between 18 and 43 years old (*M* = 21; *SD* = 3.42). 

### 2.2. Data Collection

The analysis of environmental attitudes was performed by applying a standardised validated instrument; that is, the scale of environmental attitudes towards specific problems by Moreno, Corraliza, and Ruiz [41]. This scale consists of 50 items, grouped into 10 subscales, that respond using a 4-point Likert format (1 = None, 2 = A little, 3 = Quite a lot, 4 = Completely). The standardised Cronbach’s alpha of the scale is 0.838, and the construct validity has a correlations’ absolute value of 0.116. This means that participants’ answers tend not to be random [41].

This study analysed a total of 20 items, corresponding to the subscales of pollution, biodiversity, natural spaces, and recycling. Each subscale refers to different aspects of environmental attitude:Pollution (P): it is mainly related to the attitude towards environmental pollution (P1 and P2), the belief that temperature increases due to the use of fossil fuels (P3), the social perception linked to ecological disasters (P4), and the attitude towards environmental pollution offences (P5).Biodiversity (BIO): it gathers aspects related to the concern for biodiversity preservation (BIO1 and BIO2), the extinction of natural and animal species (BIO3 and BIO4), and the attitude towards cooperating with organisations that protect the environment (BIO 5).Natural spaces (NS): it is linked to nature preservation (NS1), the attitude towards contributing financially and personally to natural spaces preservation (NS2 and NS3), the perception of the natural surface decrease (NS4), and the social perception towards the use of recycled paper (NS5).Recycling (RE): it involves the respondent’s own attitude towards the act of recycling (RE1), the social perception of recycling in the immediate area (RE2 and RE3), and the belief regarding recycling impact (RE4 and RE5).

The Cronbach alpha coefficient obtained when applying the scale in this specific context shows that it is a reliable measurement (α = 0.617). This is confirmed through Guttman’s split-half coefficient (0.611).

In parallel, the exploratory factor analysis specifies the total variance explained by the items that make up each subscale (Table 1). Likewise, this paper confirms the values are acceptable in the sampling adequacy of Kaiser Meyer Olkin (KMO), ranging from 0.512 to 0.606.

### 2.3. Data Analysis

Data collection took place during the first term of the 2018–2019 academic year. After choosing the groups for the sample, they were invited to participate in the study. Participants were provided with a link to the questionnaire, which was created online through Google Form. 

Once all the answers were collected, they were analysed by means of different data-analysis software. For statistical-descriptive data and the *t*-distribution, the analysis programme of quantitative data “SPSS”, version 24 (IBM Corp., Armonk, NY, USA), was used. Nevertheless, the correlations among the latent variables were established with the software “AMOS”, version 24 (IBM Corp., Armonk, NY, USA). The analysis with AMOS allows one to perform a confirmatory factorial analysis or SEM, depending on the purpose of the study. In this paper, as it is not about the validation of a questionnaire, we refer to SEM. This type of model shows the regression values of each item in the dimension to which it corresponds.

## 3. Results

The multivariate normality of data regarding skewness and kurtosis has allowed performing the parametric *t*-test and the determination of SEM, given that the verification of this premise is a key condition in this regard [42]. Furthermore, the values associated with skewness must be under two, and below seven for kurtosis [43].

In the different subscales, the mean is above two, which is a positive predisposition towards each environmental attitude. Nevertheless, answers are confirmed to be homogeneous, due to the fact that the standard deviation is reduced. The greatest dispersion is found within the recycling subscale, which shows greater heterogeneity in the answers. Skewness and kurtosis have appropriate values (X < 2 and X < 7) (Table 2).

In order to verify the existence of statistically significant differences based on gender, the parametric *t*-test was used in independent samples. It is evidenced that such differences based on gender do not exist in each subscale (Table 3). Nonetheless, the mean women’s environmental attitude towards biodiversity and natural spaces is slightly higher than men’s. 

The determination of SEM is a graphic representation of the correlations between subscales. Therefore, the values obtained in covariance and (*R*) correlation indicate the positive interdependence among the different subscales (Table 4). In this sense, the higher the value in each subscale’s environmental attitude, the higher the value obtained in the remaining ones. Moreover, the positive correlation between P <—> BIO (*R* = 0.89), BIO <—> NS (*R* = 0.97), NS <—> RE (*R* = 0.78), P <—> NS (*R* = 0.72), BIO <—> RE (*R* = 0.84), and P <—> RE (*R* = 0.82) is gathered.

On the other hand, the critical thinking value (CT) determines the statistical significance of data (*p*-value = X < 0.05) in those scores higher than 1.96 [44]. Concerning scores, statistically significant correlations are generated among NS <—> RE (*p*-value = ***), P <—> NS (*p*-value = ***), and P <—> RE (*p*-value = ***).

The goodness-of-fit indexes of SEM are normal and confirm the adequacy of data: the root mean squared error of approximation (RMSEA = 0.05); the goodness-of-fit index (GFI = 0.90); the root mean residual index (RMR = 0.04); the Tucker-Lewis index (TLI = 0.67); the parsimony goodness-of-fit index (PGFI = 0.70), and the comparative fit index (CFI = 0.72).

For its part, the SEM consists of the correlations between the four subscales, which relate to specific environmental attitudes: pollution, biodiversity, natural spaces, and recycling. Pollution (P) is defined through five observable variables (P1, P2, P3, P4, and P5) with regression values ranging between 0.06 and 0.58. Biodiversity (BIO) is defined through a further five observable variables (BIO1, BIO2, BIO3, BIO4, and BIO5) and presents regression values ranging between 0.09 and 0.63. Natural spaces (NS) have five observable variables (NS1, NS2, NS3, NS4 and NS5) with regression values between −0.14 and 0.49. Lastly, recycling (RE) consists of five other observable variables (RE1, RE2, RE3, RE4, and RE5) with regression values ranging between −0.18 and 0.52 (Figure 2).

## 4. Discussion

The Moreno, Corraliza, and Ruiz scale [41] has allowed us to analyse the environmental attitudes of future primary education teachers towards different issues; in particular, pollution, biodiversity, natural spaces, and recycling were addressed. Results reveal a positive predisposition towards the different subscales, which suggests that future teachers are aware of worldwide challenges caused by environmental pollution and the loss of both biodiversity and natural spaces [1,2,32,37].

In particular, the analysis of the pollution subscale (P) reflects favourable results in terms of attitudes regarding environmental pollution. The consolidated belief that temperature increases are caused by the use of fossil fuels, among other factors, [7,8] is emphasised. Moreover, future teachers have a high level of social perception, and a critical and inflexible attitude towards offences related to environmental pollution.

Concerning biodiversity preservation (BIO), the score obtained indicates trainee teachers’ concern about nature preservation [9,10,11]. Consequently, the extinction of natural and animal species is another main concern [6]. It is important to stress that most of the students identify with the work of environmental organisations and express their intention to cooperate with them.

For its part, natural spaces preservation (NS) also constitutes a key point for biodiversity preservation. Future teachers’ perceptions of the natural surface decrease are at a high level. In this sense, they have a favourable attitude towards natural spaces preservation [10]. 

As for recycling (RE), the mean obtained in this subscale is the lowest mean of all, and the one with the largest dispersion. This may be because the percentage and act of recycling in Spain is the lowest in the European Union [16]. Nevertheless, in general the attitude is positive towards recycling and the fact of recycling diligently (Seymour, 2016). In parallel, it is evidenced that future primary education teachers are highly aware of the positive impact of recycling on the environment [13,14,15]. 

Overall, primary education trainee teachers show consolidated environmental attitudes towards current environmental issues [9,12,27,28,29]. This fact is important considering that environmental education is key to responding to current environmental issues through new generations in future [20,21,22,23,24].

Furthermore, in line with previous studies [17], no statistically significant differences in environmental attitudes have been found based on gender. However, although the difference is not significant, certain distinctions are found concerning average scores for the biodiversity and natural spaces subscales in favour of women [35]. 

With respect to the correlations determined in SEM, all of them have a positive interdependence. In particular, the link between natural spaces and recycling is positive. Therefore, the higher the environmental attitudes towards natural spaces preservation, the higher the positive perception towards recycling. It works similarly regarding the attitudes towards environmental pollution and natural spaces protection [6], and regarding the environmental attitude towards pollution and the fact of having a favourable attitude towards recycling [5].

Based on the thread of research carried out on environmental attitudes, the data of this paper are aligned with the findings obtained in previous studies, which indicate a high level of environmental attitudes in students [32,33,34,35,36]. On the contrary, other studies focused on a similar population (future primary education teachers) reveal insufficient levels of environmental attitudes [30].

## 5. Conclusions

Generating environmental attitudes is essential in the fight against climate change and the reduction of environmental pollution. It is necessary to continue educating future generations on these matters, which are key for the survival of the human race. In this regard, governments must get involved and legislate to include environmental education across the board within the different educational stages [30,32].

For its part, this study has addressed the perspective of future primary education teachers, who will be responsible for educating new generations. All the questions raised have been answered as follows:

▪ What is the environmental attitude of future primary education teachers towards pollution and its repercussions for biodiversity and natural spaces?

The environmental attitude is critical. This is clearly shown by the data collected, with scores above two (X > 2). It is confirmed that future primary education teachers know that environmental pollution has a direct impact on biodiversity preservation and on the degradation of natural spaces [6,36]. 

▪ What are future primary education teachers’ attitudes towards recycling?

Their attitude is favourable (X > 2). However, this attitude has the greatest dispersion and the lowest mean; yet, considering the context and general data for recycling in Spain, our data are above the average citizen [16]. This is positive, given that these teachers are responsible for educating the citizens of tomorrow.

▪ Are there statistically significant differences in the environmental attitudes according to gender?

It is verified that there are no statistically significant differences based on gender [17]. Nevertheless, the women’s mean is slightly higher regarding those matters related to biodiversity preservation and natural spaces [35].

▪ Does the level of attitudes towards the issues related to pollution, biodiversity preservation and natural spaces have an influence on having a positive predisposition towards recycling?

Yes, it does. This premise is confirmed through SEM. In this sense, the correlation between pollution and recycling is statistically significant, as well as the correlation between natural spaces and recycling. Although the correlation between biodiversity and recycling is not significant, the same yardstick remains: a positive interdependence among subscales. Hence, if the attitudes towards any of those subscales increases, the attitudes towards recycling will also increase. 

All this definitely has an impact on the fulfilment of the research aims; that is, the analysis of future primary education teachers’ environmental attitudes, the determination of the differences based on gender, and the resolution of subscale correlations.

Among the study constraints, we can observe that its population belongs to the same university and academic year. We consider that this is more than adequate for a first-instance study, and it reflects a part of the reality in the context of where the study has been carried out. Nevertheless, it will be advisable for future analyses to include other academic years and degrees.

In line with these considerations, the following future prospects are noted: (i) continuing to research the environmental attitudes of future teachers in other universities and degrees (degree in Early Education; Master’s degree in Secondary Education Teaching); (ii) analysing environmental attitudes in other university degrees, with the aim of understanding the attitudes according to the knowledge branch; (iii) applying the Moreno, Corraliza, and Ruiz scale [41] in similar contexts to contrast the data obtained in this study.

Finally, the future of biodiversity preservation and environmental pollution is in the hands of new generations. Training future primary education teachers is key to preparing them to face environmental challenges and for them to be able to pass on their knowledge to citizens of the future [17,18,19].

## Figures and Tables

**Figure 1 ijerph-16-00362-f001:**
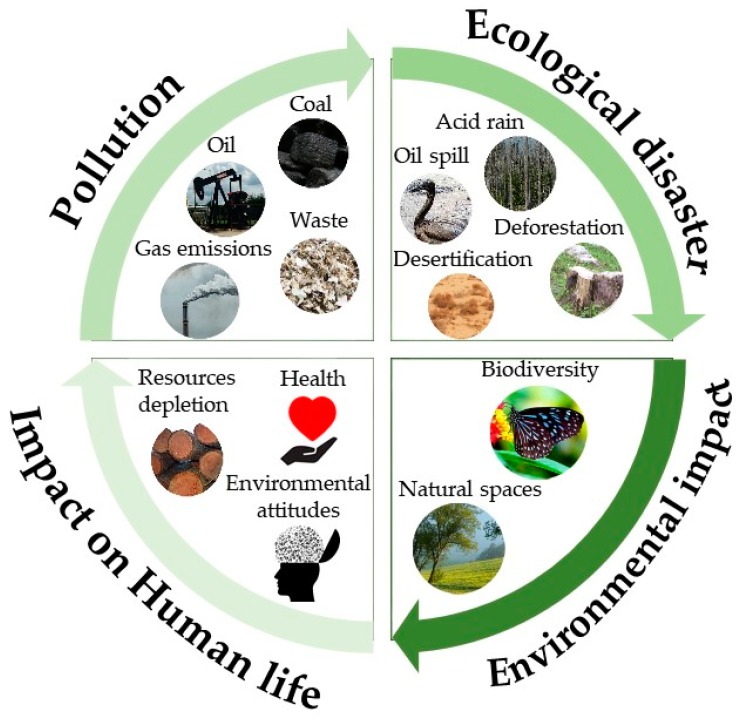
Cycle of the impact of the human being on nature.

**Figure 2 ijerph-16-00362-f002:**
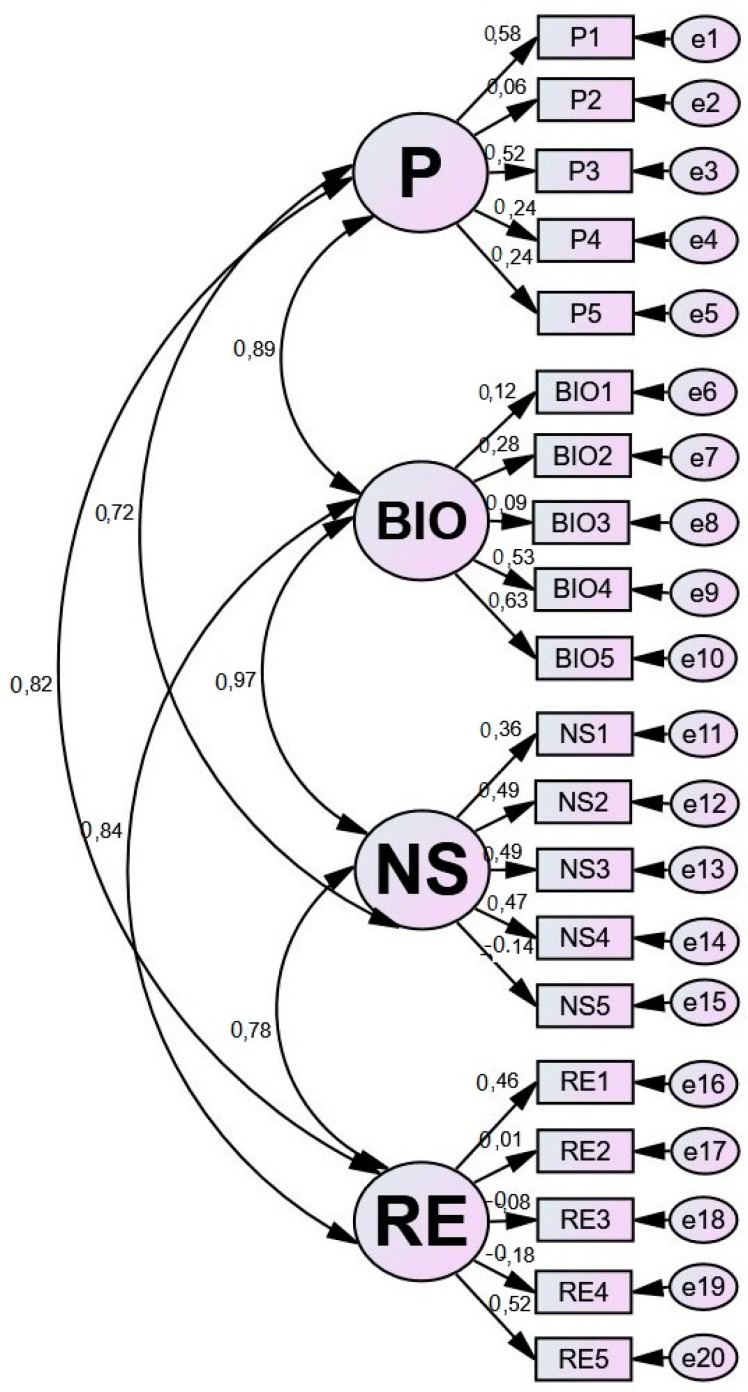
Estimations of the structural equation model. Note: Chi-square = 323,834; df = 164; *p*-value = 0.000; P = Pollution; BIO = Biodiversity; NS = Natural Spaces; RE = Recycling.

**Table 1 ijerph-16-00362-t001:** Exploratory factor analysis by subscale.

Subscale	Total Items	Variance	Total Variance	KMO	Bartlett’s Test of Sphericity		
					Chi^2^	df	*p*-Value
Pollution	5	2	50.140	0.576	49.945	10	0.000
Biodiversity	5	2	50.329	0.572	56.313	10	0.000
Natural spaces	5	2	53.240	0.606	83.715	10	0.000
Recycling	5	2	51.750	0.512	52.418	10	0.000

Note: KMO = Kaiser Meyer Olkin; df = Degrees of Freedom.

**Table 2 ijerph-16-00362-t002:** Descriptive statistics.

Subscale	Mean	SD	Skewness	Kurtosis
Pollution	3.17	0.28	−0.64	0.10
Biodiversity	3.11	0.42	−0.70	0.51
Natural spaces	3.02	0.56	−0.56	0.24
Recycling	2.80	0.63	−0.23	−0.02

Note: SD = Standard Deviation.

**Table 3 ijerph-16-00362-t003:** Test t by gender in the different subscales.

Subscale	Gender	N	Mean	SD	*t*	df	*p*-Value
Pollution	Man	79	3.17	0.23	−0.06	305	0.57
	Woman	229	3.16	0.30			
Biodiversity	Man	79	2.97	0.39	−1.99	305	0.23
	Woman	229	3.15	0.43			
Natural Spaces	Man	79	2.92	0.52	−1.35	305	0.33
	Woman	229	3.05	0.58			
Recycling	Man	79	2.80	0.52	−0.27	305	0.44
	Woman	229	2.80	0.67			

Note: SD = Standard Deviation; df = Degrees of Freedom.

**Table 4 ijerph-16-00362-t004:** Covariances and correlations.

Relation	Covariance	SE	CT	*p*-Value	*R*
P <—> BIO	0.03	0.020	1.77	0.076	0.89
BIO <—> NS	0.02	0.017	1.72	0.085	0.97
NS <—> RE	0.06	0.017	3.79	***	0.78
P <—> NS	0.06	0.016	4.07	***	0.72
BIO <—> RE	0.02	0.016	1.73	0.082	0.84
P <—> RE	0.08	0.019	4.65	***	0.82

Note: P = Pollution; BIO = Biodiversity; NS = Natural Spaces; RE = Recycling; SE = Standard Error; CT = the critical thinking value; *p*-value *** = X < 0.001.

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
