# Peer review of "Environmental Attitudes in Trainee Teachers in Primary Education. The Future of Biodiversity Preservation and Environmental Pollution"

_ijerph, 2019, doi:10.3390/ijerph16030362_

Reviewer 1 Report

in the attchment you find the reviewed paper.

Author Response

Point 1: If you take the scale of environmental attitude you can only talk about environmental attitude. Environmental awareness consists of knowledge, attitude and action.

Response 1: The commentary on this conceptualization seems appropriate. Likewise, the term environmental awareness has been modified throughout the text by environmental attitude

Point 2: Add Spain in the keywords.

Response 2: Spain has been added as a keyword.

Point 3: Please name some institutions and private companies.

Response 3: The name of some public and private companies has been added, such as Spanish Office of Climate Change, United Nations, Goldman Sachs, Google, Apple, Microsoft

Point 4: Which one? Referring to some measures. Line 35.

Response 4: It has been clarified that these measures deal with recycling and waste reduction

Point 5: Figure is strange. Health (Belongs to environmental impact). Resources (In my opinion it belongs to pollution)

Response 5: The figure has been clarified. Human impact has been changed by Impact on human life. Therefore, health belongs to this section, as well as resources. This concept it refers to the resource depletion, which has been clarified in the figure. In general, the figure represents the cycle of the impact in human life. It starts with pollution that causes the ecological disasters, this, cause at the same time an environmental impact that affects the life of the human being.

Point 6: Universities have tried to implement environmental education as cross-disciplinary in the official syllabus. Please show an example.

Response 6: Universities have tried to implement environmental education as cross-disciplinary in the official syllabus, proof of this, it was the establishment of the Network for the Curriculum's Environmental of Higher Education in Spain (ACES Network).

Point 7: Better is to study environmental action. Environmental action is even better suited as a predictive tool.

Response 7: This paper has focused on the environmental attitude. In addition, the instrument used directly measures the environmental attitude construct. So it is timely as you indicated above to talk about the term attitude.

Point 8: Why do you not interpret the means?

Response 8: The means have been interpreted in the corresponding section, which according to the guidelines for the writing of a research report is the discussion section.

Point 9: The highest difference is in the scale of biodiversity. Is there a reason for it?

Response 9: In the section corresponding to the discussion this result is addressed: Furthermore, in line with previous studies [17], no statistically significant differences in environmental attitudes have been found based on gender. However, although the difference is not significant, certain distinctions are found concerning average scores for the biodiversity and natural spaces subscales in favour of women [35]. Regarding the reason, it is not possible to establish clear ideas of why this occurs with the type of methodology used: quantitative method.

Point 10: Moderate English changes required.

Response 10: It is added a certificate of sworn official translator for English and Spanish language, authorised by the Spanish Ministry of Foreign Affaairs, European Union and Cooperation.

Reviewer 2 Report

It is an  interesting paper that needs major revisions till publishing in the journal. My comments for improving are:

-In the Introduction the first part (lines 32-63) are very general for the theme of the article. I propose to the authors to replace it with more relevance references about teacher's attitudes about environmental awareness  in different european countries.

There are also problems in the sample method:

Why the authors used only the students of the second year of education and not at the rest of the years?

Can you describe with more information about  the AMOS analysis you make in the article? 

Author Response

Point 1: In the Introduction the first part (lines 32-63) are very general for the theme of the article. I propose to the authors to replace it with more relevance references about teacher's attitudes about environmental awareness in different european countries.

Response 1: The text starts talking about the general to the particular, so we believe it is essential to start writing contextualizing it at the macro level to move to a micro level. Regarding the inclusion of references on teaching attitudes on environmental awareness in different countries, lines 77-101 contain this information that the reviewer demands. In this paragraph, different previous investigations on the topic of study are analyzed.

Point 2: There are also problems in the sample method: Why the authors used only the students of the second year of education and not at the rest of the years?

Response 2: As specified in the article, the sample corresponds to students in the second year of the grade in primary education. The calculation of the representative sample size indicates that the population can be generalized to the entire promotion. The second course has been specifically selected since it is when the student has not yet started the subjects corresponding to the experimental sciences that can interfere with their attitude towards the environment. And the first year students have not yet finished consolidating in the University, they are in their adaptation period. So the choice of the second year is not casual, responds to idyllic conditions to focus the study of environmental attitudes. Also, reviewing the literature on the topic shows similar samples, where it focuses on a specific course [30, 32, 33, 34, 35, 36]

Point 3: Can you describe with more information about the AMOS analysis you make in the article?

Response 3: It has been added this paragraph: The analysis with AMOS allows to perform a confirmatory factorial analysis or a structural equation modelling, depending on the purpose of the study. In this paper, as it is not about the validation of a questionnaire, we refer to a structural equation modelling. This type of model shows the regression values of each item in the dimension to which it corresponds. Lines 173-177.

Point 4: English language and style are fine/minor spell check required.

Response 4: It is added a certificate of sworn official translator for English and Spanish language, authorised by the Spanish Ministry of Foreign Affaairs, European Union and Cooperation.

Round  2

Reviewer 2 Report

All the necessary improvements have  done to the manuscript. The author's comments properly answer to my review. 

I suggest that the particular paper is appropriate for publication in your journal.